# Hair of the Dog: Identification of a *Cis*-Regulatory Module Predicted to Influence Canine Coat Composition

**DOI:** 10.3390/genes10050323

**Published:** 2019-04-26

**Authors:** D. Thad Whitaker, Elaine A. Ostrander

**Affiliations:** Cancer Genetics and Comparative Genomics Branch, National Human Genome Research Institute, National Institutes of Health, Bethesda, MD 20894, USA; dustin.whitaker@nih.gov

**Keywords:** *Canis lupus familiaris*, hair, coat, GWAS, association, gene regulation

## Abstract

Each domestic dog breed is characterized by a strict set of physical and behavioral characteristics by which breed members are judged and rewarded in conformation shows. One defining feature of particular interest is the coat, which is comprised of either a double- or single-layer of hair. The top coat contains coarse guard hairs and a softer undercoat, similar to that observed in wolves and assumed to be the ancestral state. The undercoat is absent in single-coated breeds which is assumed to be the derived state. We leveraged single nucleotide polymorphism (SNP) array and whole genome sequence (WGS) data to perform genome-wide association studies (GWAS), identifying a locus on chromosome (CFA) 28 which is strongly associated with coat number. Using WGS data, we identified a locus of 18.4 kilobases containing 62 significant variants within the intron of a long noncoding ribonucleic acid (lncRNA) upstream of *ADRB1*. Multiple lines of evidence highlight the locus as a potential *cis*-regulatory module. Specifically, two variants are found at high frequency in single-coated dogs and are rare in wolves, and both are predicted to affect transcription factor (TF) binding. This report is among the first to exploit WGS data for both GWAS and variant mapping to identify a breed-defining trait.

## 1. Introduction

All domestic dogs are members of the same species, *Canis lupus familiaris.* Following strong human selection, modern dogs have been divided into over 400 distinct populations termed breeds. Each breed has its own standard: a list of requisite physical and behavioral traits by which all breed members are judged. Most breeds were developed within the last 200 years [1,2,3]. Genetic analysis using breed-specific standards, rather than individual measurements, is a useful method for accurately identifying genetic variants controlling breed-specific traits such as those associated with morphology [4,5,6,7]. One notable feature which distinguishes breeds from one another is hair type. Previous reports have identified variants associated with fur length, texture, curliness, baldness, and shedding [8,9,10,11,12,13]. Some genes control the same phenotypes in other species [8,14,15,16], thus highlighting the dog as an ideal system for the identification of genetic variants controlling mammalian hair composition.

In humans, hair can be used as an expression of oneself, and changes in the overall structure can have social and mental health implications. In particular, thinning or loss of hair (alopecia) has a large psychological impact on both men and women, potentially increasing anxiety and affecting body image [17,18], thus highlighting the need to better understand features of hair composition and growth. Hair follicle composition in dogs is similar in anatomical structure to that observed on the human scalp as both are composed of a compound structure with multiple hair shafts extending from each follicle [19,20,21].

Double-coated dogs have two layers of hair: the courser primary or guard hairs which aid in the prevention of superficial injuries, repel excess moisture, and provide the primary coloration and texture patterns; and the more numerous secondary hairs or undercoat which are soft and downy in appearance and protect dogs from extreme temperatures. Wolves, the closest living ancestor to the modern domestic dog [22], also have both a primary coat and undercoat, and this is assumed to be the ancestral trait. Single-coated dog breeds only have primary hairs and thus usually shed less because the undercoat is more prone to falling out with the change of season. There are roughly equal numbers of breeds with double- and single-coats distributed throughout the 161 breeds for which extensive phylogenetic studies have been done [23]. Some breed groups, such as those of Alpine origin, have predominantly one phenotype, in this case the double coat.

This study leveraged breed standard phenotypes of domestic dog breeds with either double- or single-coats in combination with two genome-wide datasets [7,23] to identify a single, narrow locus on CFA28 contained within the intron of an uncharacterized lncRNA, upstream of *ADRB1*, which we show is strongly associated with coat number. This locus contains multiple pieces of evidence predicting a regulatory role, suggesting that it contains a *cis*-regulatory module required for the development and/or maintenance of the undercoat in single-coated dogs.

## 2. Materials and Methods 

### 2.1. Coat Number Assignment

Designation as a double- or single-coated breed was based on cross-referencing breed standards from three major sources: the American Kennel Club [24], Federation Cynologique Internationale [25], and the United Kennel Club [26]. All hairless breeds or those with potentially confounding phenotypes, such as long versus short-haired Dachshunds, were excluded from the analysis. The list of all breeds and dogs used herein, along with their coat number, is listed (Appendix A). Additional breed-specific phenotypes (shedding, average length, and furnishings) were assigned based on previous publications [8,9].

### 2.2. Sample Dataset for Single Nucleotide Polymorphism-Based Analysis 

The initial unpruned data set contained Illumina 170K Canine HD SNP array genotypes from 591 dogs representing 72 double-coated breeds and 526 dogs from 65 single-coated breeds, as listed (Appendix A) [23]. All SNP genotypes were called using CanFam3.1 genome assembly positions [27]. SNPs with ≥10% missing genotypes or a minor allele frequency (MAF) of ≤1% were pruned using PLINK v1.9.0 [28], resulting in a total of 150,132 SNPs.

### 2.3. Sample Dataset for Whole Genome Sequence-Based Analysis 

For the second analysis, we used a recently published canine WGS catalog containing approximately 91 million SNP and indel variants [7]. We filtered canines from the catalog to include only dogs with ≥10x autosomal coverage; no more than four dogs per breed; and, when possible, equal representation of males and females. All wolves with ≥10x autosomal coverage were retained. This generated a dataset composed of 141 double-coated dogs from 56 breeds, 96 single-coated dogs from 44 breeds, and 35 wolves (Appendix A). We pruned variants in the same manner as in the SNP array dataset (missing genotypes ≥10%; MAF ≤1%) generating a final dataset of approximately 15 million SNP and small indel variants. 

### 2.4. Genome Wide Association and Linkage Disequilibrium

We performed GWAS for both SNP and WGS datasets used GEMMA v0.96 [29]. An estimated centered relatedness matrix of all dogs was created using default GEMMA settings prior to performing the GWAS. The additional hair traits listed above were used as covariates in the SNP array analysis to correct for confounding hair phenotypes. The Wald test statistic was calculated for all GWAS analyses. For both GWAS, we used two thresholds: (1) the 5 × 10^−8^ standard used for human SNP associations [30]; and (2) Bonferroni-corrected significance levels based on SNP or variant number from our own data [31].

We calculated linkage disequilibrium (LD) using PLINK from the WGS single- and double-coated dog data. We calculated pairwise r^2^ between the tagging variant (chr28:24,863,224) and all other variants within the locus for all dogs. We defined the region of interest to include all variants with an r^2^ correlation ≥0.5.

LocusZoom v1.3 [32] was used to create the regional Manhattan plot of CFA28 with a custom dog database using (1) all SNP positions extracted from our 15M WGS filtered variants; and (2) the genomic information for all transcripts [33]. To use this program at this locus, we converted all variant positions and gene information from CFA28 to CFA1 to allow accurate plotting of data with gene annotations.

### 2.5. Structural Variant Prediction Analysis

Structural variants (SVs) were called using DELLY v0.7.8 [34] for 40 dogs, 20 each for single- and double-coated breeds, using their BAM files (Appendix A, column “SV”). Sites were merged by phenotype and SVs genotyped against the merged list. BCFtools v1.8 [35] was used to compile all SVs, and variants with a “PASS” filter were further considered.

CNVnator v0.3.3 [36] was used to call SVs in the same 40 BAM files. Variants were called only within CFA 28 (NC_006610.3). Program default values with a bin size of 150 base pairs were used. After calls were performed for all dogs, BCFtools was used to merge all 40 VCF files.

### 2.6. Splice Site Prediction Analysis

Prediction of alternative or cryptic splicing motifs was performed using the Alternative Splice Site Predictor (ASSP) software [37] with default parameters. For the representative single-coated dog, we extracted an input sequence of the variants plus/minus 1000 bp from the University of California, Santa Cruz Genome Browser. To represent a double-coated variant, only the single nucleotides were altered from the CanFam3.1 sequence to represent the major allele in double-coated dogs.

### 2.7. Transcription Factor Binding Prediction

Genomic regions immediately surrounding selected variants were extracted from the CanFam3.1 reference sequence on the UCSC Genome Browser [33]. The reference sequence is from DNA isolated from a Boxer, which is a single-coated dog. We extracted the variant, plus/minus 20 nucleotides. For the double-coated input sequence, we altered the sequence of single-coated dogs at the single nucleotide from derived to ancestral allele for the input sequence. Transcription factor binding prediction was performed using AliBaba v2.1 [38] as a part of the TRANSFAC suite [39]. 

## 3. Results

### 3.1. SNP Array-Based GWAS

We initially performed a SNP-based GWAS to identify loci associated with single versus double-coats in dogs. Using 526 single- and 591 double-coated dogs (Table 1), we identified six SNPs at four loci that exceed genome-wide significance (5 × 10^−8^) and seven that exceed Bonferroni significance (Figure 1A and Appendix A, Appendix A) on canine chromosomes CFA1, 2, 13, and 28 (λ = 0.9976, Figure 1A and Appendix A, Appendix A). The strongest association was on CFA28 (7.94 × 10^−16^) but contained only a single significant SNP (CFA28:24,866,296). The CFA1 and 13 loci have been previously associated with other hair phenotypes, including: length, shedding, and furnishings [8,9]. When we adjusted for each of the hair covariates, only the single SNP on the CFA28 locus remained significant (Appendix A, *p*-value 8.58 × 10^−10^). The presence of only a single significant SNP defining the CFA28 locus may be indicative of a very small haplotype, which is not surprising given the large number of dog breeds considered in the analysis.

### 3.2. Whole Genome Sequence-Based GWAS and Linkage Disequilibrium

An additional GWAS using variants from a comprehensive set of SNPs and indels selected from a recently developed dataset of 722 whole genome sequenced canids [7] was performed in order to refine the CFA28 signal. These data include 96 single- and 141 double-coated dogs (44 and 56 breeds, respectively). We identified 87 variants on three loci, including two loci on CFA1 and one locus on CFA28, that reach either genome-wide and/or Bonferroni significance (Figure 1B & Figure 1; *p* < 3.38 × 10^−9^; λ = 0.9935; Appendix A). Of the 87 variants, 74 reside on CFA28. To ensure that missing genotypes did not skew the *p*-values of candidate variants, especially given the small number of dogs in this analysis, data from all dogs with missing genotypes at any of the above 74 variants were removed. Re-analysis of the filtered GWAS (data not shown) did not result in a dramatic shift in the 10 most significant variants, six of which remain in top 10 upon re-analysis. We thus used the full dataset for subsequent analyses. Linkage disequilibrium was calculated between all variants on CFA28 and the most significantly associated variant (CFA28:24,863,224) in all dogs to define a region of LD (r^2^ ≥ 0.5). A narrow, 18.4 kb locus (CFA28:24,852,538–24,870,897) lies fully within the intron of an uncharacterized long noncoding RNA (lncRNA), upstream of the *ADRB1* gene (ENSCAFT00000056258, Figure 1C), which we termed *ADRB1-AU1* based on lncRNA naming conventions [40].

### 3.3. Genomic Structural Variants

Our initial approach toward identifying highly associated or causal variants was to look for gross structural alterations between double- and single-coated dogs at the CFA28 locus. We used two programs to predict SVs in 20 each of double- and single-coated dogs, using one dog/breed (Appendix A). However, we did not identify any SVs in any dog within the 18kb locus. We expanded our search, plus/minus one megabase (Mb), and observed 130 SVs using DELLY and 364 using CNVnator (Appendix A). Most predictions from CNVnator (81%) were detected in only a single dog, and no SV was seen in more than three individuals. Using DELLY, we observed no SVs exclusive to either single- or double-coated breeds. Most SVs (88%) were found in equal proportion between phenotypes (plus/minus two individuals, 10% of group), and none was found in more than 75% of dogs for each phenotype. These data suggest that no large SV or copy number variants explain the lack of an undercoat in single-coated dogs. 

### 3.4. Fine Mapping of Small Variants

We next sought to identify likely associated small variants in the WGS dataset for further investigation. The intronic locus in LD with the tagging variant contained 62 variants reaching Bonferroni significance (3.38 × 10^−9^). Allele frequencies for each variant were calculated for both dog groups and for wolves. We designated the wolf major and minor alleles as ancestral and derived alleles, respectively. Only variants for which the ancestral allele frequency in wolves and double-coated dogs was ≥0.5 and the frequency in single-coated dogs was ≤0.5 were retained, yielding 28 variants (Table 1). The derived allele frequency of each variant in single-coated dogs was ≥0.9. Two of the 28 variants were particularly intriguing due to their low population frequency in wolves: CFA28:24,870,184 has a derived allele frequency of 0.0571, and the derived variant at CFA28:24,860,187 is not found in any wolves (Table 1). It is worth noting here that derived allele frequencies in double-coated dogs are possibly higher than those observed in wolves as the conformation standard for some double-coated breeds lists a lack of an undercoat as a fault, although this phenotype still exists within some such breeds. Further investigation was warranted to understand these results.

### 3.5. Impact of Variants on Gene Regulation

We hypothesized that the above two intronic variants (Figure 2A) might act in the regulation of either splicing or gene expression. Using the web-based splice site predictor ASSP, we did not find that either variant was located within or would change a potential cryptic splicing donor or acceptor site (data not shown). We did observe, however, that one of the two variants (CFA28:24,860,187) is positioned approximately 600 base pairs away from a TF binding site on the canine UCSC track (Figure 2A), as indicated by the curated annotation of regulatory regions using ORegAnno [41]. ChIP-sequencing results for three histone modifications indicative of active gene enhancers (H3K4me1, H3K4me3, and H3K27ac) were also found within the CFA28 locus [42]. These two pieces of evidence suggest a regulatory role for this locus. We performed in silico analysis to determine if either variant lies within a regulatory region not currently annotated by ORegAnno and to test if either of the two SNPs could alter TF binding. We observed that both are predicted to alter binding by adjusting a highly conserved nucleotide within the TF consensus sequence (Figure 2B,C). All of these genes are expressed in the adult dog hair follicle [43] and in key hair follicle cell subtypes within the developing mouse [44,45] (Table 2). In aggregate, these data suggest that these two variants have a potential regulatory role, and this entire locus may be a cis-regulatory module.

## 4. Discussion

Six genetic variants controlling hair features including length, texture, curliness, shedding, presence of furnishings, and hairlessness have been successfully identified in the domestic dog [8,9,10,11,12,13]. In this study, we used two published genome-wide datasets to investigate the phenotype, a SNP array of 150,132 variants and 15 million WGS variants [7,23]. We observed a single locus on CFA28 that is strongly associated with the breed trait of presence or absence of an undercoat. Our use of the latter dataset provides an early adoption of WGS data for performing ultra-dense GWAS in canines. The variants identified here could be used in breeding programs to select dogs for the desired coat type. For instance, the standards for many double-coated breeds list the lack of an undercoat as a serious fault. We envision that fanciers of these breeds could select for sires or dams likely to produce only double-coated puppies. Conversely, the undercoat is the predominant coat to shed, so there may be selection for single-coated dogs in pet owners, particularly those with mixed breed dogs.

The resulting locus of 18.4kb lies within the intron of *ADRB1-AU1*. Because a WGS dataset was used, we were able to easily sample all variants within the region and develop a priority list of those with significant associations. Two variants were particularly provocative as both are absent or rare in wolves, the nearest ancestor to the modern domestic dog. The low level of the derived allele in wolves could indicate the age of the variant, supported by the very small haplotype between dog breeds, or may be a sporadic variant in a hotspot of genomic instability. Similar observations have been made for derived alleles at low frequencies in wolves [46]. In silico analysis of the two derived alleles predicts that both lie within one or more consensus sequences associated with three TFs: *AP-1*, *CEBPA*, and *POU2F1*. Interestingly, all three have been shown to control expression of a gene, *KRTAP6-1,* which is involved in hair fiber diameter and curvature through both positive and negative gene regulation [47]. Three of the four genes contributing to the above TFs are expressed within the adult dog hair follicle [43], and all are expressed in mouse hair follicle precursors or component cell types [44,45]. Perhaps these TFs regulate gene expression of one or more neighboring genes, likely *ADRB1-AU1* and/or *ADRB1*. Due to the proximity of both genes to this locus, it would be reasonable to predict they are targets for gene regulation. We failed to detect either gene in a public dog hair dataset or in testes tissues (data not shown), possibly because the hair was of not of correct type (undercoat) or the relevant genes may not be expressed in testes. We nevertheless hypothesize that one or both genes plays a role in the development and/or maintenance of the undercoat hair follicles.

The role of *ADRB1-AU1* is not well defined as it is only recently annotated in dogs [48], but many other experimentally tested lncRNAs control their immediate neighboring genes [49]. It is plausible that *ADRB1-AU1* regulates the expression of *ADRB1*. No hair phenotypes have been reported in the loss of function *Adrb1* mouse, as most are embryonically lethal [50]. Interestingly, however, the *ADRB1* protein directly binds to the G-coupled protein receptor Gαs, also known as *GNAS* [51]. *GNAS* and protein kinase A (*PKA*) signaling within the epidermis tightly control hair follicle stem cell populations and an increase or decrease in this signaling results in progressive hair loss though either hair follicle stem cell exhaustion or lack of differentiation into hair follicle progenitors [52]. It is also possible that a similar mis-regulation of the *ADRB1-GNAS*-*PKA* signaling pathway in single-coated dogs leads to a depletion of mature undercoat hair follicles. Future studies will be required to define the mechanism and timing of *ADRB1* in the canine hair follicle. The dog may also serve as a system for developing treatments to maintain or recover the undercoat, highlighting dogs as a model for understanding hair thinning and loss in humans.

## 5. Conclusions

The domestic dog has been an excellent genetic model for understanding a diverse array of morphological features, including the identification of causal variants for six hair phenotypes. Here, two independent genome-wide datasets identify a locus on canine chromosome 28 that is strongly associated with the presence or absence of an undercoat in domestic breeds. Multiple lines of evidence predict that this locus is a *cis*-regulatory module. Two single nucleotide alterations in the intron of *ADRB1-AU1* are strongly associated with an undercoat in single-coated breeds, and both may play a role in controlling gene expression, likely of *ADRB1*. Current literature links *ADRB1*’s interacting proteins, *GNAS* and *PKA*, with progressive hair loss, so perhaps this pathway is also perturbed in single-coated dogs leading to an absence of the undercoat. The current study, with the identification of two likely causal SNP variants, raises the total number of hair features described in domestic dogs to seven and highlights the value of the dog for studying breed-specific phenotypes of interest in human biology.

## Figures and Tables

**Figure 1 genes-10-00323-f001:**
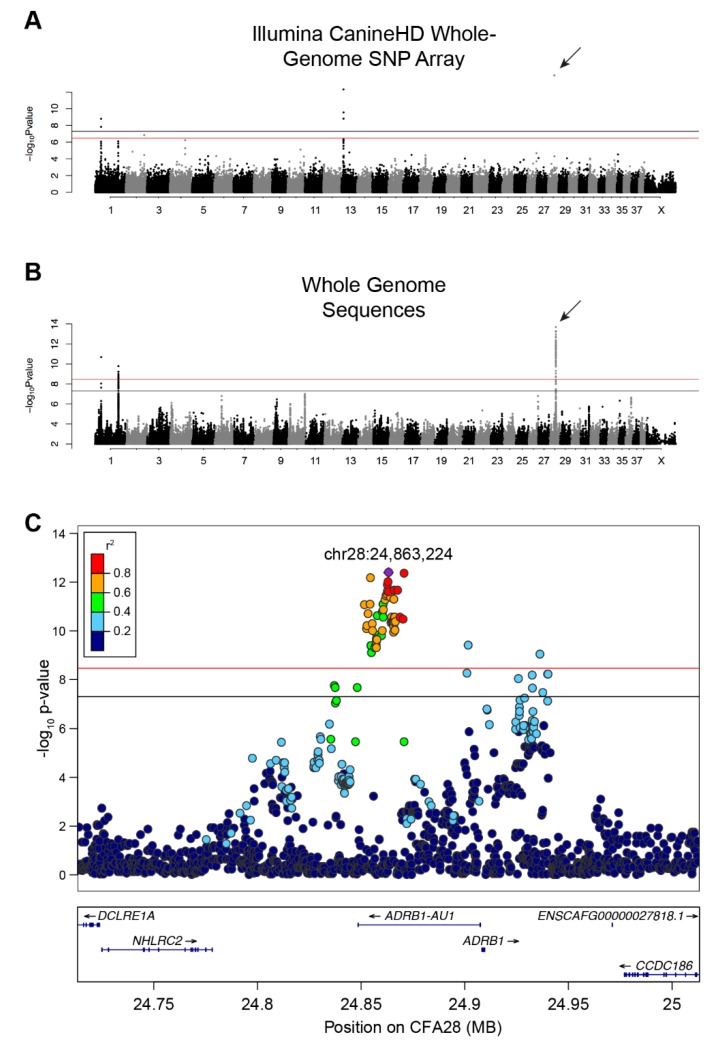
GWAS identifies a strong association on CFA28 with lack of an undercoat. (**A**) Manhattan plot of -log_10_ transformed Wald *p*-values for the SNP association of single- versus double-coated dogs. Black horizontal line indicates genome-wide significance (5.0 × 10^−8^). Red line indicates Bonferroni-corrected genome-wide significance (3.3 × 10^−7^). Four loci surpass the significance threshold, with the most associated locus located on CFA28 and represented by a single SNP (arrow). (**B**) Manhattan plot of -log_10_ transformed Wald *p*-values for the WGS association of single- versus double-coated dogs. Black horizontal line indicates genome-wide significance (5.0 × 10^−8^). Red line indicates Bonferroni-corrected genome-wide significance (3.4 × 10^-9^). Three loci, two on CFA1 and one on CFA28, exceed both thresholds. Only SNPs with *p*-value ≤ 0.1 were included in this plot. (**C**) Regional Manhattan plot of CFA28 locus from the WGS GWAS. Pairwise linkage (r^2^) of each variant was calculated relative to the most significant variant: chr28:24,863,224 (purple). All strongly correlated variants (r^2^ ≥ 0.6) with significant *p*-values reside within the intron of an uncharacterized lncRNA.

**Figure 2 genes-10-00323-f002:**
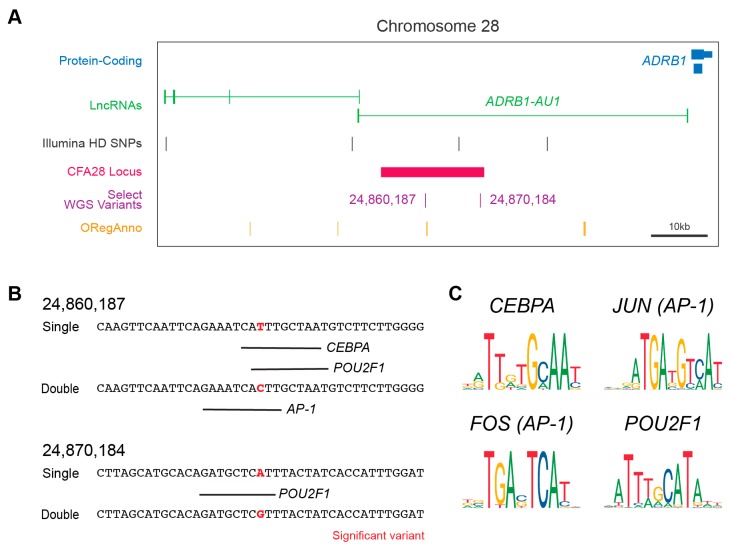
The CFA28 locus and variants effects on TF binding. (**A**) UCSC Genome Browser display centered around the CFA28 GWAS locus. The locus (black bar) is in the intron of *ADRB1*-*AU1* (green) upstream of *ADRB1* (blue). Both variants are indicated within the region. Also shown are the curated annotations from ORegAnno (orange). (**B**) In silico prediction of transcription factor binding surround the two derived alleles in wolf with low frequency. Transcription factor prediction was performed using AliBaba2 at the significant variant plus/minus 20bp in single- (derived) and double-coated (ancestral) alleles. Binding is denoted by a black line below target sequence. (**C**) The consensus sequence for each of the four transcription factors identified in (B) are shown. *AP-1* is composed of a dimer of *FOS* and *JUN*. The two variant positions lie within well conserved regions of each transcription factor.

**Table 1 genes-10-00323-t001:** Top filtered variants associated with loss of undercoat in WGS GWAS.

CFA28 Position	Ancestral Allele	Derived Allele	Wolf Derived	Double Derived	Single Derived	Wald *p*-Value *
**24,860,187**	C	T	0.0000	0.3731	0.91011	1.01 × 10^−10^
**24,870,184**	G	A	0.0571	0.3769	0.91573	3.37 × 10^−11^
24,851,582	G	C	0.1429	0.3918	0.9382	8.79 × 10^−12^
24,865,942	T	C	0.2286	0.3657	0.92135	2.23 × 10^−12^
24,862,630	A	G	0.2714	0.3657	0.91011	1.29 × 10^−12^
24,864,369	T	G	0.2857	0.4403	0.9382	4.75 × 10^−11^
24,866,307	T	G	0.3000	0.4403	0.9382	9.35 × 10^−11^
24,863,224	G	A	0.3143	0.3619	0.91011	4.04 × 10^−13^
24,865,980	C	T	0.3143	0.4366	0.9382	2.77 × 10^−11^
24,864,985	T	C	0.3429	0.444	0.94382	4.37 × 10^−11^
24,865,626	A	G	0.3429	0.444	0.9382	1.15 × 10^−10^
24,865,654	G	T	0.3571	0.4403	0.9382	6.42 × 10^−11^
24,852,776	A	AAGTCTTCAT	0.3714	0.4142	0.9382	6.11 × 10^−11^
24,853,325	T	TA	0.3714	0.3993	0.9382	1.99 × 10^−11^
24,866,155	C	A	0.3714	0.4366	0.9382	2.77 × 10^−11^
24,866,296	T	C	0.3714	0.4403	0.9382	9.35 × 10^−11^
24,865,525	A	G	0.3857	0.4403	0.9382	4.67 × 10^−11^
24,865,800	A	G	0.3857	0.4366	0.9382	4.45 × 10^−11^
24,866,114	C	A	0.3857	0.4366	0.9382	2.77 × 10^−11^
24,866,181	C	G	0.3857	0.4366	0.9382	2.77 × 10^−11^
24,866,484	G	A	0.3857	0.4478	0.9382	4.28 × 10^−11^
24,867,588	C	T	0.3857	0.3657	0.92135	2.23 × 10^−12^
24,854,456	A	ATTTGT	0.4143	0.3993	0.94944	6.77 × 10^−13^
24,864,154	A	T	0.4286	0.4403	0.9382	4.23 × 10^−12^
24,864,658	A	G	0.4286	0.4366	0.9382	2.77 × 10^−11^
24,863,186	C	CT	0.4429	0.4067	0.90449	2.54 × 10^−12^
24,865,062	A	AACAAC	0.4429	0.4403	0.9382	4.67 × 10^−11^
24,868,735	G	A	0.4429	0.3843	0.92135	2.87 × 10^−11^

* *p*-value derived from double- versus single-coated WGS GWAS. Bolded values indicate two variants for which in silico TF binding analysis was run.

**Table 2 genes-10-00323-t002:** Expression levels of transcription factors in various hair cell types.

	Dog	E14.5 Mouse		P5 Mouse		
Gene Name	Hair Follicle	Placode	Dermal Condensate	Bulge Stem Cells	Transit Amplifying Cells	Dermal Papilla
*CEBPA*	32.9	22.6	2.5	87.1	16.4	38.2
*FOS*	37.5	4.2	6.1	444.2	237.9	425.0
*JUN*	88.0	7.7	17.6	374.5	286.4	797.1
*POU2F1*	0.42	14.3	6.5	12.9	14.3	9.5

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
