# Peer review of "Hair of the Dog: Identification of a Cis-Regulatory Module Predicted to Influence Canine Coat Composition"

_genes, 2019, doi:10.3390/genes10050323_

Round 1

Reviewer 1 Report

The authors perform across-breed genome wide association studies for canine coat composition using SNPs from Illumina arrays and variants called from whole genome sequencing data. They identify a single, 18kb region of association lying within the intron of a lncRNA and posit that two SNPs exert a regulatory effect on ADRB1. The manuscript is exceptionally well written and the methods are appropriate and thorough. I only have a few minor comments:

On Line 217 the text states that one of the candidate SNPs is 600 kb from a transcription factor binding site. On Line 222, it is suggested that one of the SNPs might disrupt a transcription factor binding site. This initially seems contradictory. It would be helpful to state that the in silico search was performed to find previously uncharacterized TF binding sites.

I would like to see a discussion of the implications of your findings. I’m curious if, based on your results, you think this phenotype is governed by just one locus? I think I might have expected, given the large size of these association studies, a lower P-value? Also, I assume based on the filtering strategy that there is allelic variation within breeds, which also suggests there may be additional loci involved. Along these lines, how do you think the wolves might have acquired the SC allele? Introgression? Could they be wolf-dog hybrids? Or do you think the allele is old enough to have originated in wolves?  

Fig 2A: It is difficult to follow the rows on Figure 2A, I think because they are different heights. And one of the bars on the OregAnno track looks out of line.

Line 16: confirmation should be conformation

Author Response

The authors perform across-breed genome wide association studies for canine coat composition using SNPs from Illumina arrays and variants called from whole genome sequencing data. They identify a single, 18kb region of association lying within the intron of a lncRNA and posit that two SNPs exert a regulatory effect on ADRB1. The manuscript is exceptionally well written, and the methods are appropriate and thorough. I only have a few minor comments:

We sincerely thank you for your review and for your appreciation of the work performed here.

On Line 217 the text states that one of the candidate SNPs is 600 kb from a transcription factor binding site. On Line 222, it is suggested that one of the SNPs might disrupt a transcription factor binding site. This initially seems contradictory. It would be helpful to state that the in silico search was performed to find previously uncharacterized TF binding sites.

Thank you for your comment regarding transcription factor binding. We have adjusted this according to your recommendation to help clarify our findings.

I would like to see a discussion of the implications of your findings. I’m curious if, based on your results, you think this phenotype is governed by just one locus? I think I might have expected, given the large size of these association studies, a lower P-value? Also, I assume based on the filtering strategy that there is allelic variation within breeds, which also suggests there may be additional loci involved. Along these lines, how do you think the wolves might have acquired the SC allele? Introgression? Could they be wolf-dog hybrids? Or do you think the allele is old enough to have originated in wolves?

We have added to our Discussion the potential and practical implications of our data.

We cannot rule out additional loci in controlling single- versus double-coats. Possibly one of the CFA1 loci that were found to regulate shedding potential co-regulates this phenotype.

You are correct in pointing out that there is allelic variation within breeds. Nineteen breed standards for double-coated dogs indicate that having a single coat is a major fault. To us this indicates that both coat types could be seen in these breeds, though likely the single-coated individuals are less common. This is a downside of using breed standards over individual phenotypes and likely why our p-values might not be as significant as could be expected.

We mention a couple of possible reasons why the rare derived variant is present in wolves, though any statements are speculative without further study.

Fig 2A: It is difficult to follow the rows on Figure 2A, I think because they are different heights. And one of the bars on the OregAnno track looks out of line.

For Figure 2A, we originally used a screenshot of the UCSC Genome Browser to display annotations within this region. We have redrawn the window with better spacing and color coordinated annotations for improved clarity. Thanks for your comment; the new figure looks much neater.

Line 16: confirmation should be conformation

Thank you for catching our typo. Fixed.

Reviewer 2 Report

line 86 typo: analysis 

fig 1 A: 7th SNP on chromosome 2 (or 3?): this chromosome is not mentioned on line 142

fig 2: I see A and (a), B and (b) but no C for (c)?? 

check for double-spaces

I would like to read on the practical implication of this finding in dog breeding (MGT? - selection for full coats in all dogs??) and for human health (cfr. psychological impact of hair variation as mentioned in introduction). 

Author Response

line 86 typo: analysis

Thank you for catching this typo. Fixed.

fig 1 A: 7th SNP on chromosome 2 (or 3?): this chromosome is not mentioned on line 142

We have corrected this mistake in our results section. We did not mention CFA2 by name even though acknowledged its SNP when discussing the number of significant markers (and it was listed in the supplementary table).

fig 2: I see A and (a), B and (b) but no C for (c)??

Thank you for catching this error on our part. We have added “C” to the figure.

check for double-spaces

We did identify a couple of rouge double spaces. Thank you for pointing this out.

I would like to read on the practical implication of this finding in dog breeding (MGT? - selection for full coats in all dogs??) and for human health (cfr. psychological impact of hair variation as mentioned in introduction).

We have added information into our discussion about the practical implications of these data on dog breeding and management. We also expanded on the next step required for understanding the impact on human health.